# MAxSIM: multi-angle-crossing structured illumination microscopy with height-controlled mirror for 3D topological mapping of live cells

Pedro Felipe Gardeazabal Rodriguez[1], Yigal Lilach[2], Abhijit Ambegaonkar[3], Teresa Vitali[1], Haani Jafri[4], Hae Won Sohn[3], Matthew Dalva ![ORCID][4,6], Susan Pierce[3] & Inhee Chung ![ORCID][1,5✉]

Mapping 3D plasma membrane topology in live cells can bring unprecedented insights into cell biology. Widefield-based super-resolution methods such as 3D-structured illumination microscopy (3D-SIM) can achieve twice the axial ( ~ 300 nm) and lateral ( ~ 100 nm) resolution of widefield microscopy in real time in live cells. However, twice-resolution enhancement cannot sufficiently visualize nanoscale fine structures of the plasma membrane. Axial interferometry methods including fluorescence light interference contrast microscopy and its derivatives (e.g., scanning angle interference microscopy) can determine nanoscale axial locations of proteins on and near the plasma membrane. Thus, by combining super-resolution lateral imaging of 2D-SIM with axial interferometry, we developed multi-angle-crossing structured illumination microscopy (MAxSIM) to generate multiple incident angles by fast, optoelectronic creation of diffraction patterns. Axial localization accuracy can be enhanced by placing cells on a bottom glass substrate, locating a custom height-controlled mirror (HCM) at a fixed axial position above the glass substrate, and optimizing the height reconstruction algorithm for noisy experimental data. The HCM also enables imaging of both the apical and basal surfaces of a cell. MAxSIM with HCM offers high-fidelity nanoscale 3D topological mapping of cell plasma membranes with near-real-time ( ~ 0.5 Hz) imaging of live cells and 3D single-molecule tracking.

[1] Department of Anatomy and Cell Biology, George Washington University, School of Medicine and Health Sciences, Washington, DC, USA.
[2] Nanofabrication and Imaging Center, George Washington University, Washington, DC, USA. [3] Laboratory of Immunogenetics, National Institute of Allergy and Infectious Disease, National Institutes of Health, Rockville, MD, USA. [4] Department of Neuroscience, Thomas Jefferson University, Philadelphia, PA, USA. [5] Department of Biomedical Engineering, GW School of Engineering and Applied Science, George Washington University, Washington, DC, USA. [6] Present address: Department of Cell and Molecular Biology, Tulane University, New Orleans, LA, USA. ✉email: inheec@gwu.edu

Biochemical and cell biological processes such as signaling and cell adhesion result from tightly orchestrated interactions of cell surface proteins within the complex plasma membrane (PM) environment[1-3]. The PM constantly reshapes itself into dynamic 3D structures with nanoscale topology[4-6]. These topological changes can alter the distribution of cell surface proteins, markedly affecting interactions between proteins and lipids and thus influencing biological outcomes[7-9]. Therefore, a truly mechanistic understanding of biological processes at the PM requires imaging techniques that can visualize 3D nanoscale interactions of PM components such as proteins and lipids. Moreover, real-time visualization of PM topological changes in live cells will inform the dynamic nature of these nanoscale interactions[8-12].

Widefield-based super-resolution methods such as 3D-structured illumination microscopy (3D-SIM) can achieve twice the axial ( ~ 300 nm) and lateral ( ~ 100 nm) resolution of widefield microscopy in real time in live cells[13]. This resolution improvement has enabled monitoring of various plasma membrane events and sub-cellular organelles in real time. However, twice-resolution enhancement cannot sufficiently visualize nanoscale fine structures of the plasma membrane. Fluorescence interference contrast microscopy (FLIC) is a widefield microscopy technique that incorporates optical interferometry to perform nanometer-scale axial localization[14,15]. FLIC creates an axial interference pattern along the excitation beam path due to self-interference of an incident beam with its reflection off a silicon (Si) mirror covered with a step-wise patterned silica (SiO$_2$) layer. Theoretically, FLIC can offer axial localization information of a thin object such as the PM, but it cannot localize thicker objects with extensive placement of chromophores in the axial direction. While FLIC has some limited applications for axial localization of the PM, it requires uniform chromophore labeling of the sample surface and a sample size that spans multiple micron-sized SiO$_2$ oxide steps.

Subsequent FLIC derivative methods with varying incidence angles[16-20], such as scanning angle interference microscopy (SAIM), do not have such constraints and do not require SiO$_2$ patterning to achieve nanoscale topological mapping[16,19]. SAIM can be used for axial localization in live cells at ~ 0.3 Hz[19]. Lateral imaging is diffraction-limited and most SAIM applications have been primarily used to map the topology of the basal cell surface, focal adhesion sites, and cytoskeletons underneath the basal cell surface that was adhered to the SiO$_2$/Si mirror[19,21-24]. There have been fewer attempts to map the apical cell surface on the SiO2/Si mirror, probably due to the high background incorporated into the signal and increased fitting uncertainty.

To enable robust topological mapping of both basal and apical cell surfaces, while achieving excellent height reconstruction fidelity and time resolution, as well as super-resolution lateral imaging, we developed multi-angle-crossing structured illumination microscopy (MAxSIM) with a height-controlled mirror (HCM) and a substantially improved nonlinear-least-square-based fitting algorithm. Instead of placing cells only on the SiO$_2$/Si mirror, cells can also be located on the bottom of the glass substrate with a custom-fabricated HCM (a standard 1-μm-thick SiO$_2$-covered Si mirror with a ridge structure) that is located at a specific distance above the cells (Fig. 1a, b). The ridge height of the HCM enables a preliminary estimation of the initial height parameter, which is the most crucial factor for fitting fidelity. The HCM also allows users to select an optimal ridge height for a given cell type to further improve height reconstruction fidelity. The HCM is reusable, thus saving the time and costs required for fabrication (See the HCM cleaning procedure in the method section). Our vastly optimized fitting algorithm overcomes the challenges of fitting noisy raw data to the theoretical formula by determining the best initial height parameter and optimal sub-

angle ranges for fitting. The optoelectronic control of varying incidence angles is used both for axial interferometry and 2D-SIM, thus the super-resolution lateral imaging is enabled. All of these improvements offered by our MAxSIM platform enable 3D topology mapping of live cells in near-real-time ( ~ 0.5 Hz), combined with 3D single-molecule tracking, which we showcase by imaging the apical and basal surfaces of fixed and live cells of diverse types.

## Results

**MAxSIM with HCM enables high-fidelity axial localization.** The basis of MAxSIM is a custom SIM system inspired by *fast*SIM[25] to generate excitation beams at multiple incident angles to create incident angle-dependent axial interference patterns with the presence of an HCM along the optical axis (Fig. 1a). SIM is a widefield microscopy technique that breaks the diffraction limit by patterning excitation light beams[13,26-28]. As previously done by various groups[25,28-30], we used a spatial light modulator (SLM; 2048 × 1536 pixels) that creates a grid pattern in the light path for diffraction. Diffracted beams from the SLM become *s*-polarized by the azimuthal linear polarizer (Fig. 1a; Supplementary Fig. 1a) as in *fast*SIM[25]. Subsequently, one or two beams are selected using Fourier filters[27] (Supplementary Fig. 1b) installed in a filter wheel located at the conjugate plane of the objective backfocal plane for MAxSIM. The high-speed filter wheel (filter switching time <30 ms) allows different imaging modes on the same cells. For instance, selecting the ±1$^{st}$-order diffraction beams enables 2D-SIM imaging of the same cells, even with the presence of an HCM, since the lateral interference of the symmetric ±1$^{st}$-order beams remains intact with the HCM (Supplementary Fig. 2). The selected *s*-polarized diffraction beams form a grating pattern at the sample plane through interference. Separation distance between the two beams for 2D-SIM was determined for optimal excitation numerical aperture.

In axial interferometry with an HCM, a one *s*-polarized +1$^{st}$-order beam was chosen to create axial light interference fringes with its reflection off the Si surface in the HCM (Fig. 1a). The HCM was fabricated using a high-quality Si mirror covered with a standard 1-μm-thick SiO$_2$ layer[19] to produce ridges (height: 5-25 μm) via lithography (Fig. 1b). Different SLM grid patterns were used to generate light beams with different incident angles at the objective tip (Fig. 1c). Our calibration data demonstrates excellent accuracy, as the incident angles are within a remarkable 2% margin of error compared to the theoretical angles (Supplementary Fig. 3). We did not select two-beam axial interferometry for MAxSIM using both ±1$^{st}$-order beams as in 2D-SIM, because the incidence-angle-dependent axial intensity patterns varied laterally (Supplementary Fig. 4), complicating height reconstruction. In the one-beam scenario, axial interference was laterally constant (Fig. 1d, top). In the two-beam geometry with symmetric incident light beams relative to the optical axis with the presence of a mirror, the normalized lateral interference pattern was the same regardless of axial positions (Supplementary Fig. 2). Thus, 2D-SIM is still enabled in the MAxSIM hardware configuration. At the same time, 3D-SIM is not feasible with an HCM, due to altered lateral and axial interference patterns by the presence of the 0$^{th}$-order beam. However, this optical condition can enhance the axial resolution of 3D-SIM[31]. As used in SAIM[19], MAxSIM generates an incident angle-dependent fluorescence intensity plot that varies axially. This fluorescence intensity modulation, which contains axial location information for a chromophore, is approximately proportional to the excitation interference fringe pattern, as assumed and validated in previous studies[14,19,32].

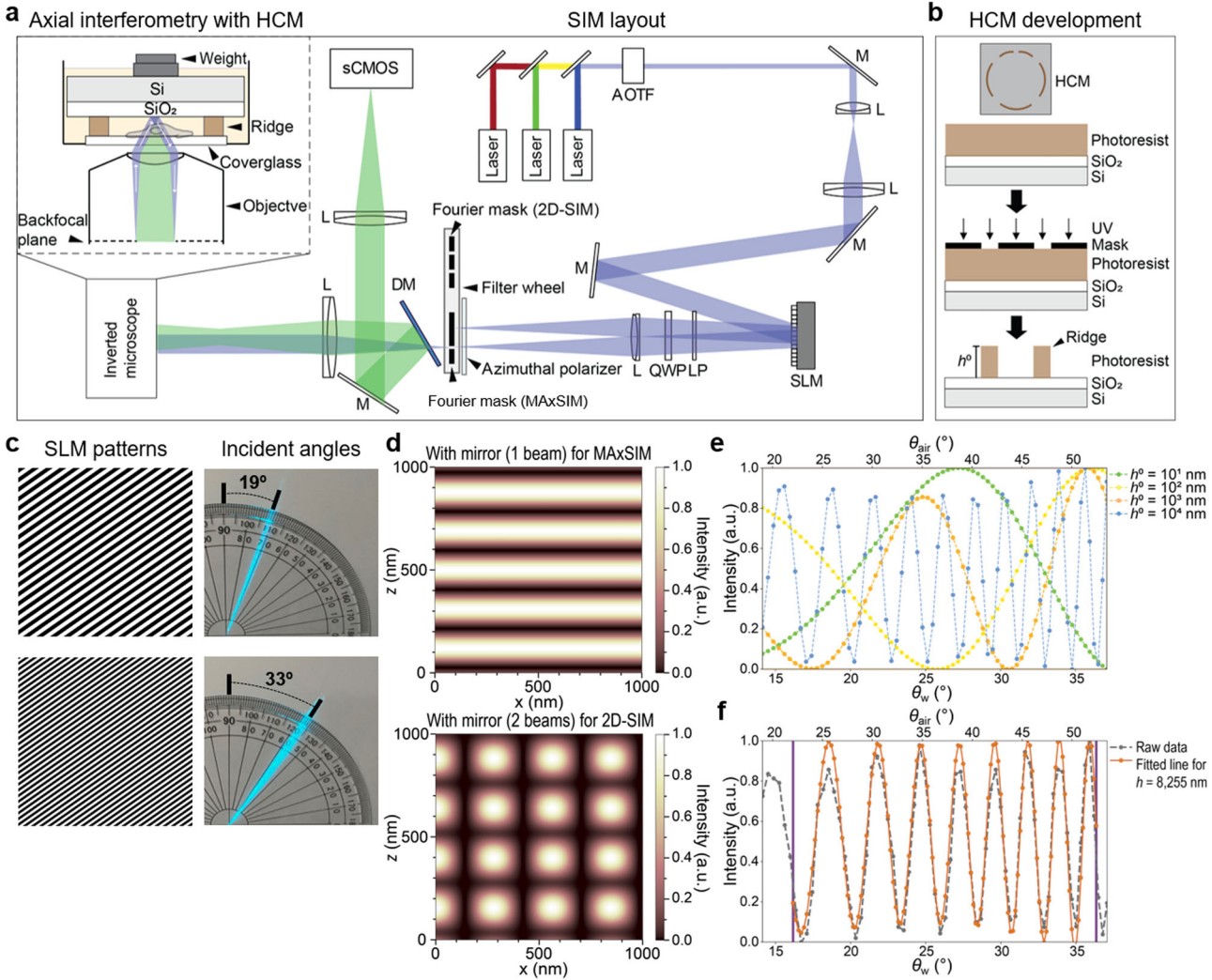

**Fig. 1 Optomechanically controlled MAxSIM with HCM. a** Optical layout of MAxSIM. AOTF: acousto-optic tunable filter; M: mirror; L: lens; SLM: spatial light modulator; LP: linear polarizer; QWP: quarter-wave plate; DM: dichroic mirror; Si: silicon wafer; SiO$_2$: silicon dioxide layer; sCMOS: scientific complementary metal–oxide–semiconductor. **b** Schematic (top: birds-eye view showing patterned ridge structure on SiO$_2$/Si layer) of the HCM fabrication procedure using a negative photoresist mask (bottom). The ridge structure is asymmetric between the top and bottom to define sample orientation. Ridge height is denoted as $h^0$ (bottom). **c** Example diffraction patterns (left) uploaded on the SLM to achieve incident angle ($\theta_{air}$ = 19° and 33° in air) as measured using a protractor in the air above the objective (right) with a 488-nm laser. **d** Simulated excitation light ($\lambda$ = 488 nm) intensity patterns in x-z dimensions with the presence of a mirror using one beam excitation with incident angle $\theta_{air}$ = 19° (top, MAxSIM) and two symmetric beams with incident angle $\theta_{air}$ = 60° (bottom, 2D-SIM). **e** Simulated light intensity modulations, $I(\theta, h)$, using one-beam excitation as a function of the incident angle in the air ($\theta_{air}$, top x-axis) and in water immersion medium ($\theta_w$, bottom x-axis) at axial locations at the ridge tips, $h^0$ = 10, 100, 1000, and 10,000 nm away from the HCM. The number of interference fringes increases at a higher position from the mirror. **f** Measured incident angle-dependent fluorescence light intensity modulation of Alexa 488 dye on the bottom glass substrate with an HCM ($h^0$ = ~10.7 μm) and excitation at $\lambda$ = 488 nm, which is normalized between (0, 1) (gray dashed line). The fitting range of the best performance was selected between purple vertical bars by our reconstruction algorithm and the height ($h$) = 8255 nm was retrieved with <1% fitting uncertainty (NELD = 0.008) between the gray dashed line (raw data) and fitted orange solid line.

We fit the raw data to the theoretical excitation interference fringe pattern formula[14] (Eq. 1) using the non-linear least square algorithm (Levenberg–Marquardt method)[33], which we further refined and optimized to improve reconstruction fidelity.

$$I = I_0|1 + r_{eff}^{TE}(\theta, h)e^{i\Phi(\theta,h)}|^2 \qquad (1)$$

$I$ is the excitation intensity variation; $I_0$ is a constant value; $r_{eff}^{TE}$ is the Fresnel reflection coefficient; and $\Phi$ is the phase difference between incident and reflected beams in the medium at $h$ below SiO$_2$[14] (for more details on the theoretical background, see Supplementary Figs. 5 and 6). Fig. 1e shows simulated examples

of the excitation intensity modulation at different heights ($h$) with varying incident angles.

The number of interference fringes depends on the SiO$_2$ thickness as well as the height of a chromophore, as demonstrated by our simulated data (Supplementary Fig. 7). Different SiO$_2$ thicknesses such as 500 nm[34] or 10 μm (both are commercially available) yield similar (in the 500 nm case) or more excitation interference fringes (in the 10 μm case) than for the 1 μm thick SiO$_2$ case[27]. An adequate number of modulation fringes within an incident angle range is crucial to yield high-fidelity height reconstruction. For instance, $h$ < 1000 nm empirically leads to poor reconstruction when using a Si mirror covered with a 1-μm-thick SiO$_2$ layer, as demonstrated by the example raw data and

fitted curves for 100 nm microspheres directly placed on a SiO2/Si mirror (Supplementary Fig. 8). A chromophore distance >5 μm away from the 1-μm-thick SiO$_2$-covered Si mirror produces many fluorescence interference fringes, enabling higher-fidelity height reconstruction using our algorithm (Supplementary Fig. 9), compared to cases with shorter distances (<1 μm). This reinforces the advantage of using an HCM with an optimal ridge height positioned above cells located on a bottom glass substrate, instead of placing cells directly on the SiO$_2$/Si mirror as used in SAIM[19]. Additionally, the HCM ridge enables precise vertical placement of the mirror on the optical axis, which cannot be achieved by the previous scheme[19,34] in which a weight is placed on the cell-plated SiO$_2$/Si mirror to prevent floating. Our method also enables the custom selection of ridge height, facilitating high-fidelity height reconstruction tailored to specific cell types (as different cells have different cell heights).

To determine reconstruction fidelity by evaluating fitting uncertainty, we devised a new metric called normalized extrema location difference (NELD) to assess deviations between theoretical extrema and observed extrema in incident angle-dependent fluorescence intensity curves. The deviation between theoretical and observed extrema positions was larger with a greater peak (+) or valley (−) width. Thus, we weighted the deviation by the width of the corresponding peak or valley. The equation for NELD (Eq. 2) is thus:

$$\text{NELD} = \frac{1}{n' + m'} \left( \sum_{j=2}^{n+m-1} \frac{\sqrt{(\theta_{o_j^{+(-)}} - \theta_{e_j^{+(-)}})^2}}{(\theta_{o_{j+1}^{-(+)}} - \theta_{o_{j-1}^{-(+)}})} \right) \quad (2)$$

$m$ and $n$ are the total numbers of maxima and minima, respectively, and $m'$ and $n'$ are the total numbers of peaks and valleys, respectively, in incident angle-dependent fluorescence intensity curves within an angle range; $\theta_{o(e)_j^+}$ and $\theta_{o(e)_j^-}$ are incident angle positions that correspond to intensity maxima (+) and minima (−), respectively, in observed (o) or expected/theoretical (e) incident angle-dependent intensity curves; and $(\theta_{o_{j+1}^{-(+)}} - \theta_{o_{j-1}^{-(+)}})$ is the width of an observed peak or valley. A peak or a valley is determined if a minimum or maximum location is situated between two adjacent maxima or minima. Our reconstruction algorithm determines the sub-incident angle range for fitting that is associated with the lowest NELD value (Supplementary Fig. 6). It is worth pointing out that our algorithm is optimized for fitting data with at least 4 interference fringes within the given angle range. Thus, it may not yield high-fidelity fitting for fluorescent objects placed on the SiO2/Si mirrors due to an insufficient number of interference fringes. In such cases, it is recommended to use the original Levenberg-Marquardt least square fitting algorithm instead.

We used a glass substrate spin-coated with IgG1-Alexa 488 conjugates as a control to take MAxSIM images. One example is shown in Supplementary Fig. 10. The raw MAxSIM data plot at one pixel point demonstrates a high-fidelity fit (NELD < 0.1), yielding an estimated height of 8,255 nm (Fig. 1f). As expected, the overall height distribution from three independent experiments display a narrow Gaussian width of 77 ± 35 nm (mean ± standard deviation), as shown in Supplementary Fig. 11. Experimentally, optimal MAxSIM reconstruction for the incidence angle range $\theta_{air}$ = (19°, 53°) required axial location of a chromophore 5–25 μm away from the bottom SiO$_2$ layer of the HCM to produce sufficient yet not excessive fluorescence intensity modulation fringes within the incidence angle range. To validate that this height range indeed led to high-fidelity nanoscale axial localization, we theoretically calculated axial localization accuracy of an object height of 5–25 μm by generating simulated intensity curves and fitting them using our fitting

algorithm. We indeed found excellent localization accuracy (~0.7%) (Supplementary Fig. 12) as our algorithm determines the initial height parameter ($h'$) that is close to the actual height of an object and identifies the optimal sub-angle window for fitting. We found that random assignment of $h'$ led to notably poor fit (Supplementary Fig. 13). To circumvent this, our algorithm determines an optimal $h'$ through iteration by selecting the initial height parameter that minimizes the NELD value around either the ridge height $h^0$ or a height value that yields the same numbers of intensity minima and maxima as those obtained from experimental intensity curves (Supplementary Fig. 6).

All of these optimization schemes, such as determining the optimal $h'$ parameter and sub-angle ranges for fitting, led to high-fidelity height reconstruction (NELD < 0.1 at each pixel point; NELD = 0.2 is an empirically determined upper cut-off for high-fidelity fitting) as shown in Supplementary Fig. 10, validating the robust nanoscale 3D topological mapping capability of our MAxSIM/HCM/reconstruction algorithm. Our Python-based height reconstruction code can be downloaded from our GitHub site. Detailed background information on our MAxSIM reconstruction algorithm is described in Supplementary Fig. 6.

**Nanoscale 3D PM topology mapping by MAxSIM with HCM.** We tested the applicability of MAxSIM by probing the surface morphology of the apical or basal surfaces of various fixed cells placed on the bottom glass substrate. The light geometry relative to the chromophore for a cell located on the bottom glass is illustrated in Supplementary Fig. 14. To validate the accuracy of PM morphology assessment through MAxSIM, we chose cell pairs exhibiting distinct PM morphologies and compared their height reconstruction images. For this purpose, we used the MCF7 and SKBR3 breast cancer cell lines, which we previously studied and found to exhibit contrasting PM morphologies corresponding to their HER2 receptor tyrosine kinase expression levels[12]. The MCF7 cell line represents relatively flat PM (normal HER2 expression), while the SKBR3 cell line can display deformed PM morphologies (associated with HER2 overexpression). We chose HCM ridge heights of ~11 and 22 μm to cover MCF7 and SKBR3 cells seeded on the bottom glass substrate, respectively. We recommend to use a HCM that is taller than the cell height to ensure that the mirror is positioned above the measured cells to perform MAxSIM on both basal (Fig. 2a, b) and apical (Fig. 2c, d) cell surface. We compared the Gaussian widths of the height distributions between the MCF7 cells (width: 784 nm) and the SKBR3 cells (width: 1410 nm) in Fig. 2. As predicted, the average widths for three MCF7 cells were 493 ± 315 nm (n = 3), which is smaller than that for three SKBR3 cells, 1009 ± 594 nm (n = 3) (Supplementary Fig. 11).

Naive B cells and germinal center (GC) B cells differ in their intrinsic antigen affinity thresholds for activation that correlate with the cellular architecture of their PM-expressed B cell receptors (BCRs)[35]. When placed on antigen-containing planar lipid bilayers, naive B cells form flat contacts with the bilayer and show uniform distribution of BCRs. In contrast, GC B cells form actin-rich, pod-like structures that concentrate BCRs at tips that contact the bilayer. Consistent with this observation, the basal surface of naive B cells (stained for BCRs) was flatter (Gaussian width: 427 nm; Fig. 2e) than the basal surface of GC B cells (1702 nm; Fig. 2f), which showed protrusions that may facilitate antigen-driven selection as part of the immune response, enhancing affinity discrimination of antigen[35]. We noted heterogeneity in the plasma membrane morphologies and height distributions of both naive B and GC B cells, potentially due to variations in local antigen densities in the lipid bilayer. Therefore,

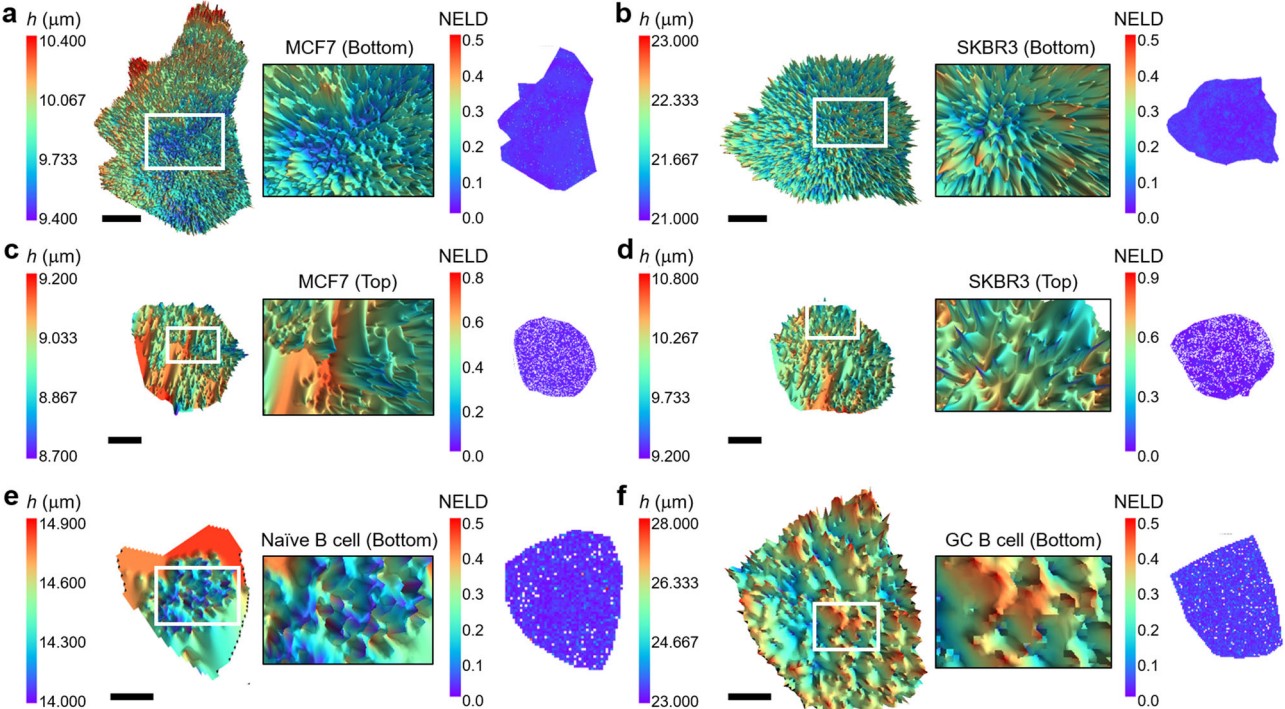

**Fig. 2 3D cell surface morphology mapping using MAxSIM with HCM. a–f** 3D topology images and zoomed-in images of specific areas (indicated by white boxes) of fixed cell samples are reconstructed from the raw MAxSIM images obtained by scanning the incidence angle (19°, 53°) with a 0.5° step size. 2D NELD maps demonstrate overall high fitting fidelity, with most values smaller than 0.1. MCF7 cells with low HER2 expression (**a**: basal; **c**: apical) and SKBR3 cells overexpressing HER2 with deformed membranes (**b**: basal; **d**: apical) were stained with wheat germ agglutinin-Alexa 555 conjugates (WGA-555). The basal cell surfaces of a naive B cell **e** and a germinal center (GC) B cell **f** with pod-like structures labeled for B cell receptor with Dylight 550-conjugated Fab fragments of antibodies against IgG or IgM heavy chain. Height scales were determined to include fitted heights with NELD < 0.1. A median filter with a kernel size = 2 was applied to reduce noise in **a–f**. Exposure time = 200 ms **a–f**. Scale bars = 5 μm **a–d** and 2 μm **e**, **f**.

we did not conduct statistical comparisons. The differential MAxSIM cell surface topologies of displayed MCF7 and naive B cells versus SKBR3 and GC B cells, respectively, align with previous results[12,35] and thus broadly validate our method.

The diffraction-limited lateral imaging of protein distribution on the cell surface limits accurate localization within the 3D PM. Thus, we integrated the 2D-SIM capability of our MAxSIM platform for lateral imaging, and combined it with MAxSIM's axial topology mapping using two distinct chromophores. We visualized protein distribution of GluA2, the obligate subunit of the AMPA receptor, in the bottom membrane of neuronal dendrites using two-color MAxSIM imaging (Fig. 3a). AMPA receptors can be located in synaptic sites and are responsible for most glutamatergic signaling in the brain. GluA2 (Fig. 3b) was found concentrated in local areas, colocalizing with wheat germ agglutinin (WGA)-stained membrane protrusions in the dendrite and in flat areas. In fact, GluA2 is reported to form nanomodules within dendritic spines and forms non-synaptic nanoclusters in flat cell areas when imaged using STED[36,37]. Thus, rather than overlaying the laterally diffraction-limited MAxSIM image of GluA2 (Fig. 3d) to the MAxSIM image of WGA (Fig. 3c), we employed super-resolution 2D-SIM imaging with an excitation numerical aperture of 1.15 (Fig. 3e, f) on GluA2 to generate a mask (Fig. 3g) to map height reconstruction solely from the masked area. The overlaid 3D image of WGA and masked GluA2 clearly showed a few sub-micron sized foci of GluA2 located within and outside PM protrusions, implying GluA2 forms nanoclusters throughout the PM of the dendrite (Fig. 3h), which is not explicitly evident in the diffraction-limited MAxSIM image overlay (Fig. 3c). This example demonstrates how the super-resolution lateral imaging capability of MAxSIM can enhance

the 3D mapping of cell surface protein distributions by more accurately defining lateral protein localizations.

The versatility of MAxSIM in creating custom interference patterns for excitation beams allows users to create custom imaging routines incorporating various modes such as axial localization by MAxSIM, 2D-SIM, and nanometer-scale lateral localization by single-molecule tracking. MAxSIM requires scanning of incident angles and thus is time-intensive, even when using SLM-based fast imaging of each frame. To achieve higher time resolution without compromising localization accuracy for near-real-time imaging of live cells, we modestly reduced the angle range [19°, 53°] previously determined for axial localization in fixed cells. Having a large scanning angle range is favored for fixed-cell imaging because it allows better height reconstruction via algorithm-based selection of an optimal sub-angle range associated with the lowest NELD value for fitting. However, for live-cell imaging, it is desirable to narrow the scanning angle range to improve time resolution while maintaining overall high reconstruction fidelity.

Experimentally, we determined a suitable range [19°, 35°] for live-cell imaging with MAxSIM. Shorter angle ranges can lead to poorer height determination fidelity. Some of the pixels that cannot be assigned to heights can be interpolated using the nearest neighboring pixels that were assigned with heights with sufficiently high-fidelity reconstruction (NELD < 0.2). Imaging was performed on live, WGA-555-stained SKBR3 cells in full growth medium in a temperature-, humidity-, and $CO_2$-controlled chamber. Using a 50-ms exposure per frame, we achieved 1.9 s per MAxSIM topology image (Supplementary Video 1). The comparable overall NELD values in the initial ($t = 0$ s) (Fig. 4a) and final ($t = 20.9$ s) (Fig. 4b) topology maps

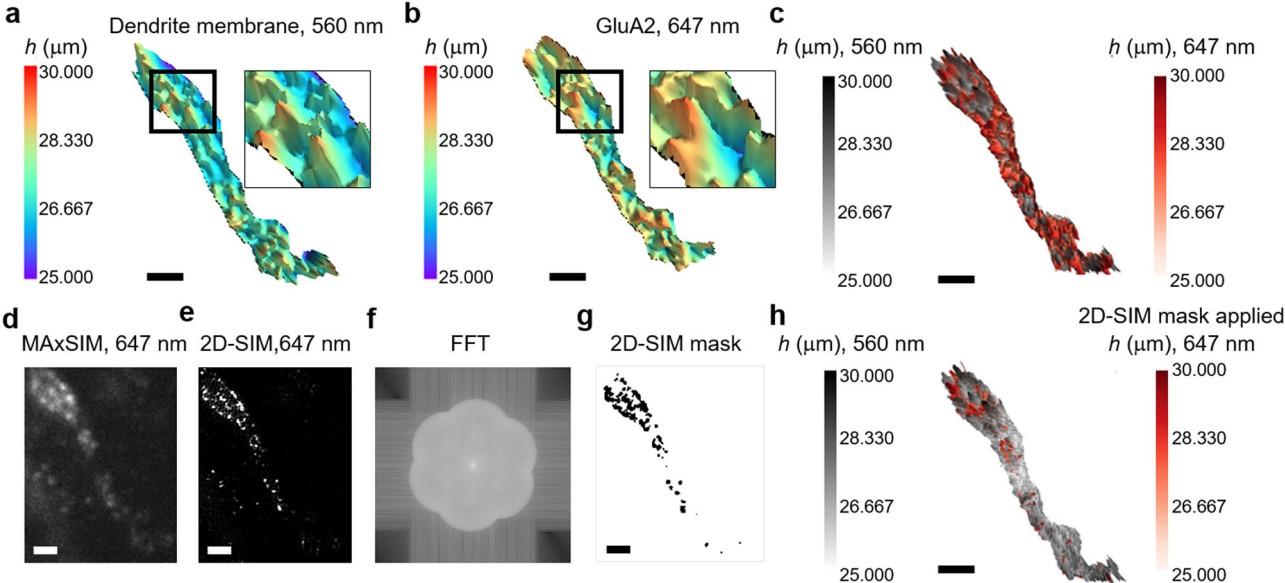

**Fig. 3 3D cell surface protein localization by MAxSIM combined with 2D-SIM imaging-based spatial mask. a, b** Reconstructed MAxSIM 3D topology and zoomed-in images (marked by black boxes) of fixed neuronal dendrites obtained from scanning the incidence angle (19°, 53°) with a 0.5° step size. Primary cortical cultures from Long Evans rat embryos stained with **a** WGA conjugated to Alexa555 (WGA-555) with laser excitation at 560 nm or with **b** anti-GluA2 antibody visualized with a secondary Atto647N antibody with excitation at 647 nm. Exposure time = 200 ms. **c**. Merged 3D topology image of WGA (gray) and GluA2 (red). **d–g**. Diffraction-limited lateral image **d** and super-resolution 2D-SIM image with excitation numerical aperture = 1.15 **e** of GluA2 on the dendrite. A fast Fourier transform image was generated (**f**), and intensity thresholding was applied to create a mask **g** from the 2D-SIM image **e**. **h** Merged 3D topology image of WGA (gray) and GluA2 (red) incorporating the mask **g**. A median filter with kernel size = 3 was applied to reduce noise **a-c**, **h**. Scale bars = 2 μm **a–e**, **g**, **h**.

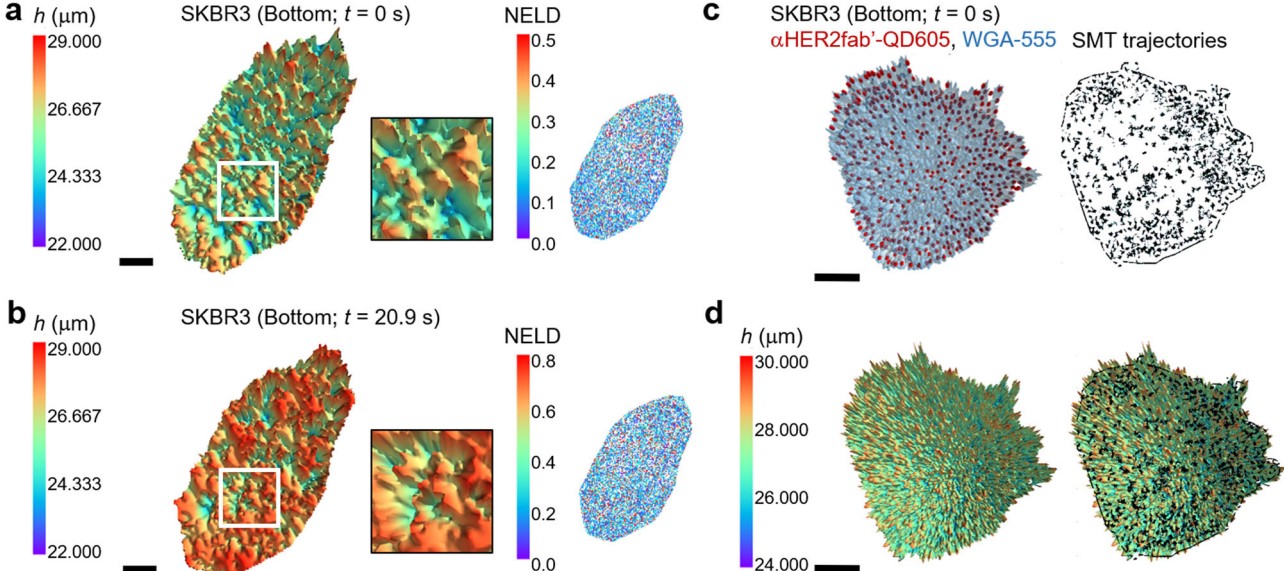

**Fig. 4 Live-cell topological mapping and 3D single-molecule tracking by MAxSIM. a, b** Live-cell 3D basal surface morphology mapping of SKBR3 cells at 1.9 s per topology map, equivalent to 0.52 Hz. The incident angle range was (19°, 35°) with 0.5° step size, and a 50-ms exposure time was used (Supplementary Video 1). The images show 3D topology, zoomed-in views of specific regions marked by white boxes, and 2D NELD images at time 0 **a** and 20.9 s **b**. **c** A 3D topology snapshot of single-molecule tracking movie of αHER2 Fab':QD605 conjugate-labeled HER2 and WGA-555 in the basal cell surface of live SKBR3 cells, while both chromophores were simultaneously excited at 560 nm. Single-molecule tracking was obtained from all 68 frames used for angle scanning, which was required to reconstruct one height map. The incident angle range was (19°, 53°) with 0.5° step size, and 50-ms exposure time was used (Supplementary Video 2). Summed trajectories of HER2 single molecules are shown in black lines. **d** A snapshot 3D topology PM map (at $t = 0$ s; Supplementary Video 3) using WGA-555 staining overlaid with summed trajectories of HER2 single molecules (black lines). A median filter with kernel size of 2 was applied to reduce noise **a–d**. Scale bars = 10 μm **a–d**.

indicate minimal bleaching over ~ 20-s acquisition time, signifying high-fidelity MAxSIM height reconstruction in live cells (See temporal progression of the raw fluorescence images and NELD ( < 0.2) values for ~ 21 s in Supplementary Fig. 15). Future efforts should be made to further improve time resolution for live-cell imaging while maintaining excellent fitting fidelity by utilizing more photostable and brighter chromophores, such as Janelia Farm dyes[38,39], and fine-tuning SiO2 thickness.

The live-cell and near-real-time imaging capability of MAxSIM prompted us to extend our technique to 3D single-molecule tracking. For simultaneous detection of 3D membrane topology and 3D single-molecule locations, we leveraged the photophysical properties of quantum dots (QDs), whose absorption is more a continuum at the higher energy of the first absorption peak[40]. We used the same excitation at 560 nm for WGA-555 and QD605 for MAxSIM and single molecule tracking, respectively, by continuously generating 68 images every 3.3 s from scanning the incidence angle range [19°, 53°] with 0.5° step (Supplementary Video 2). All 68 frames were used to track QDs laterally, and we could achieve ~20 Hz single-molecule tracking with $z$ locations obtained every 3.3 s. These results successfully demonstrate the versatility of MAxSIM for 3D topological mapping of live cells (Supplementary Video 3) and 3D single-molecule tracking (Supplementary Video 2).

## Discussion

The development of various super-resolution optical techniques[13,27,28,37,41–53] has enabled cellular imaging with a greatly improved axial resolution. For instance, widefield-based super-resolution methods such as 3D-structured illumination microscopy (3D-SIM)[13,27] can achieve twice the axial resolution ( ~ 300 nm) of widefield microscopy by creating 3D interference patterns for sample excitation. Stimulated emission depletion (STED) microscopy can yield six-fold improved axial resolution ( ~ 100 nm) by narrowing the point spread function using a bottle-shaped STED beam[54]. Further, 4Pi[42] and I[5]M[53] microscopy attain seven times the axial resolution ( ~ 100 nm) of widefield microscopy by generating axial interference patterns using two opposing objectives. 3D localization-based microscopy[55], such as interferometric PALM (iPALM)[46] (localization accuracy <20 nm), 3D-STORM[48] (20–30 nm), and point spread function engineering methods[51,52] (10–20 nm), also perform axial localization with high accuracy. Supplementary Table 1 summarizes the resolution of 3D localization microscopy methods and 3D super-resolution imaging techniques in comparison to the theoretical axial accuracy and lateral resolution of MAxSIM.

Our MAxSIM with an HCM uses SLM-based optoelectronic control of incident angles to generate angle-dependent fluorescence image data and can incorporate 2D-SIM functionality; our fine-tuned reconstruction software enables high-fidelity height reconstruction of MAxSIM images. Together, the hardware and software MAxSIM/HCM platform allows for 3D nanoscale cell surface morphological mapping and also live-cell application that can achieve near-real-time ( ~ 0.5 Hz) 3D topology mapping and 3D single-molecule tracking. It also enables super-resolution lateral imaging by enabling 2D-SIM. Employing an HCM greatly enhances the reconstruction fidelity of MAxSIM by allowing more accurate prediction of the initial height parameter, which is essential for fitting raw intensity data to the theoretical curve, and permitting versatile sample placement and reusability of the HCM. To assess height localization accuracy, we defined a new metric called NELD that measures extrema position differences between raw intensity data and theoretical curves. The NELD metric for assessing localization accuracy is superior to using the actual shapes of intensity curves, which can vary depending on experimental conditions such as inclusion of background signals. Our examples demonstrate that the MAxSIM platform can be applied for diverse research questions, including revealing unprecedented mechanistic information about critical cellular processes by providing robust details about how 3D PM topologies influence membrane protein functions and interactions with other cellular components. Obtaining real-time 3D topology information of live cells can provide new insights into the regulation and deregulation of crucial cellular processes, thereby influencing diverse fields of life sciences and clinical research.

## Methods

**Cell culture.** Cell lines (MCF7, HTB-22; SKBR3, HTB-30) that were authenticated were purchased from ATCC. Cells with low passage numbers ( < 15) were used. We confirmed that cells were mycoplasma-free via PCR analysis. MCF7 cells were maintained in DMEM with 10% fetal bovine serum and 1% penicillin-streptomycin and cultured in 5% CO2 at 37 °C. SKBR3 cells were maintained in DMEM with 10% fetal bovine serum and 1% L-glutamine in the same conditions. Cell culture conditions were kept constant during imaging experiments and throughout immunofluorescence cell preparation procedures. For imaging, cells were plated on glass-bottom dishes with 14-mm diameter (MatTek; glass thickness: No. 1.5).

**αHER2 Fab':QD conjugation.** Fab' fragments of the HER2 antibody [7C2.v2.2.LA clone; obtained via MTA (OR-216587) from Genentech] were conjugated to QD605 according to our published protocol[56], with minor modifications. HER2 antibody storage buffer was replaced with 0.1 M acetate buffer/0.01 M EDTA pH 4 through a desalting spin column (Thermo Fisher Scientific). The digestion was initiated by adding pepsin at a 40:1 (antibody:enzyme) ratio for 2 h at 37 °C. To block pepsin digestion, the reaction mixture was transferred to PBS buffer through a desalting spin column (Thermo Fisher Scientific). Then, 1.75 μL of freshly dissolved 20 mM sulfo-SMCC (Thermo Fisher Scientific) in DMSO was added to 62.5 μL of an 8 μM stock solution of amino-PEG-QD605 (Life Technologies). The mixture was incubated at room temperature (RT) for 1 h. In parallel, 100 μg of Fab' fragments were diluted in 300 μL PBS and reduced by adding 1.25 μl of 1 M cysteamine water solution at RT for 10 min. Sulfo-SMCC-derivatized QDs were separated from excess unreacted sulfo-SMCC by passing the solution over NAP-5 desalting columns (GE Healthcare) pre-equilibrated in 50 mM HEPES pH 7.2/150 mM NaCl. Similarly, reduced Fab' fragments were passed over NAP-5 columns pre-equilibrated in the same HEPES/NaCl buffer. Derivatized QDs and reduced Fab' fragments were mixed and allowed to react at RT for 2 h. Fab':QD conjugates were concentrated by ultrafiltration (Pierce Protein Concentrators, Thermo Fisher Scientific) and separated from unconjugated Fab' fragments by gel filtration (Superdex G200, GE Healthcare). Concentration of the final conjugates in PBS was calculated from absorbance at 600 nm using the extinction coefficient of QD605: 650,000 M$^{-1}$cm$^{-1}$ at 595–605 nm. Glycerol was added to the conjugate sample at a final concentration of 50% for storage at 4 °C.

**HCM fabrication with ring mask.** To prepare the ring mask, 110 nm of chromium was sputter-coated on a glass slide (25 mm × 25 mm) using a sputter (CHA criterion). The slide was coated with ~200 nm poly(methyl methacrylate). The ring pattern (see Fig. 1b) was written on the slide using an electron beam lithography system (Raith Voyager) with 50 kV acceleration

voltage and 8 nA current. The developed pattern was transferred to the chromium by wet etch for 60 s.

For HCM fabrication using the ring mask, a silicon wafer covered with 1 μm of thermal $SiO_2$ was cut into 35 mm × 35 mm chips. Chips were sonicated in isopropyl alcohol for 2 min and then dried with nitrogen. Chips were spin-coated with SU-8 2005 or SU-8 2025 photoresist at spin speeds of 2000–6000 rpm to achieve a film thickness of 5–30 μm. Chips were baked on a hot plate at 65 °C for 3 min and 95 °C for 9 min and then exposed to UV for 20 s (~200 mJ/cm$^2$) under a ring mask, followed by another post-bake at 65 °C for 2 min and 95 °C for 7 min. Samples were developed with SU-8 developer for 7 min and washed with isopropyl alcohol. Samples were heated to 120 °C for 10 min to smooth the SU-8.

A setscrew (Thorlabs, SS25S050V), with a nut (N25S0440) as a weight (Fig. 1a) was used to secure the HCM in place on top of cells in a medium.

**HCM cleaning procedure**. The HCM is reusable if the following cleaning steps are adhered to. 1. Dunk the used HCM in fresh 100% isopropanol for 10 min. 2. Rinse it with deionized water for 2 min. 3. Air dry the HCM. 4. Store it in a dust-free storage box. Our group has used the same HCM for over 2 years. The photoresist we use, SU-8, is known for its remarkable resistance (https://kayakuam.com/products/su-8-series-and-kmpr-plasma-removal-rework/), which contributes to the durability of our HCM.

**MAxSIM setup**. The custom SIM microscope was built on a Zeiss Axio Observer inverted microscope platform with an ASI motorized stage. A Zeiss C-apochromat 63 × 1.2NA W Korr UV-VIS-IR water objective (WD = 0.17 mm) was used for both MAxSIM and SIM. We tested different objectives, each varying in working distances (WD = 0.15-0.4 mm at D = 0.17 mm), immersion media (water and glycerin), and numerical aperture (NA = 1.15-1.3). After thorough evaluation, we identified that a 63x water objective with 1.2 NA (Objective C-Apochromat 63x/1.20 W Corr M27 WD = 0.28 mm at D = 0.17 mm) was the most suitable for MAxSIM. This particular objective allowed for a large excitation scan range and provided sufficient photon collection for our purposes. Although a glycerin objective with similar capabilities (63x, NA = 1.3, WD = 0.17 mm at D = 0.17 mm) could also be used, we ultimately chose the water immersion objective, because of our observation that glycerin sometimes failed to uniformly cover the objective tip, leading to non-reproducible incident angles. To ensure successful implementation of MAxSIM, it is crucial to carefully select an objective by evaluating the accuracy and precision of the incident angles in the desired scan range using different SLM patterns. The minimum requirements for an objective to perform MAxSIM are 63x, NA = 1.2 (water), and WD > 0.15 mm. VSIM, an open-source software developed by HHMI, was used to control SIM electronics. Three-color (488, 560, and 647 nm) MAxSIM and 2D-SIM imaging with an HCM were enabled. Sequential excitation at the two wavelengths or sequential use of the Fourier filter was enabled by two software-controlled filter wheels (Finger Lakes Instrumentation). Excitation grating patterns at each wavelength were generated by SLM (Forth Dimension Display)[13]. Fluorescence images were collected using a sCMOS camera (Hamamatsu Flash 4.0) with an exposure time of 30 ms. 2D reconstruction of the raw data was performed using custom software[13].

**Cancer cell imaging**. SKBR3 and MCF7 cells were stained with 1 μg/mL WGA conjugated with Alexa 555 in PBS for 2 min at RT immediately after fixation with 4% paraformaldehyde for 10 min.

For single-molecule tracking, 0.2 nM Fab'-QD conjugates were used to label individual HER2 proteins in live SKBR3 cells in a glass bottom dish for 10 min. Cells were washed twice and incubated with warm (37 °C) media before being placed in an environmental chamber at the MAxSIM microscope sample stage.

**B cell preparation**

*Tonsillar B cell isolation and preparation of B cell subpopulations*. Fresh human tonsils were obtained from patients undergoing tonsillectomies at the pathology department of the Children's National Medical Center, Washington, DC. The use of these tonsils for this study was exempted from review by the NIH Institutional Review Board, following the guidelines issued by the Office of Human Research Protections. Tonsil cells were sorted into naive B cells and GC B cells based on the expression of CD19, IgD, and CD10 as follows: naive: CD19 +, IgD +, CD10-; GC: CD19 +, IgD-, CD10 +.

*Surface BCR labeling and activation on antigen-bound PLB*. Sorted cells were incubated in 5% CO2 at 37 °C for 2 h. After incubation, cells were labeled for BCRs using Fab fragments of antibodies against IgG or IgM heavy chain (109-586-129; 109-587-043, Jackson Immunoresearch lab). Cells were then activated for 8 minutes at 37 °C using antigens attached to the planar lipid bilayer prepared[57].

For the preparation of the antigen-bound planar lipid bilayer, biotin-containing small unilamellar vesicles (SUVs) were prepared by mixing a 100:1 molar ratio of 1,2-Dioleoyl-sn-Glycero-3-phosphocholine (DOPC) and 1,2-Dioleoyl-sn-Glycero-3-phosphoethanolamine-cap-biotin (DOPE-cap-biotin) (Avanti Polar Lipids, Alabaster, AL), sonicating, and resuspending in PBS at a lipid concentration of 5 mM. Aggregated liposomes were cleared by ultracentrifugation and filtering.

Planar lipid bilayers (PLBs) were formed in Lab-Tek Chambers (Nalge Nunc, Rochester, NY), with the coverglasses replaced with nanostrip-washed coverslips. The coverslips were incubated with 0.1 μM SUVs in PBS for 10 min. After washing with 20 ml PBS, the bilayer was incubated with 2.5 mg/ml of streptavidin for 10 min, and excess streptavidin was removed by washing with 20 ml PBS. The bilayers were further incubated for 20 min with 10 nM biotinylated pAb against human Igk light chain as a surrogate for the antigen. The quality of PLBs and the mobility of antigens in the lipid bilayers were confirmed by analyzing the proteins labeled with fluorescent dyes. Activated cells were fixed with 4% paraformaldehyde for 10 min at room temperature, washed with PBS, and then chambers were packed with parafilm for courier.

**Neuron cell preparation**. Primary cortical cultures from E17-E18 Long Evans rat embryos were transfected with Lipofectamine 2000 (Invitrogen 11668019) at in vitro day 17 with pFUGW-eGFP. We complied with all the ethics regulations and disclose that the organization, IACUC, has approved the protocol. At in vitro day 24, cells were incubated with 1:250 anti-GluA2 (Sigma Millipore AB397) at 37 °C for 15 min. After 5 min of anti-GluA2 incubation, WGA-Alexa 555 (Invitrogen W32464) was added for 10 min. After the total 15 min incubation, cells were washed once with warm artificial cerebrospinal fluid. Cells were fixed with 4% paraformaldehyde/2% sucrose/0.000375% glutaraldehyde for 8 min at 37 °C. Cells were washed once with artificial cerebrospinal fluid and then treated with 0.001% $NaBH_4$ on ice for 15 min. Cells were washed three times with artificial cerebrospinal fluid and then incubated with 1:400 anti-IgG2a-Atto647N antibody (Rockland 610-156-041) in 1% ovalbumin/

0.2% cold water fish gelatin blocking buffer at RT for 1 h. Cells were washed three times with artificial cerebrospinal fluid, and the dish was filled with ~5 mL artificial cerebrospinal fluid for imaging and storage.

**Statistics and reproducibility**. Measurements of the incident angle, as shown in Supplementary Fig. 3 ($n = 2$), were conducted on two separate days as independent repeats of the same test. MAxSIM imaging was carried out on three distinct samples ($n = 3$) of the same type, with data reconstruction performed using our custom MAxSIM reconstruction algorithm.

**Reporting summary**. Further information on research design is available in the Nature Portfolio Reporting Summary linked to this article.

## Data availability
The raw data used to produce Supplementary Figs. 3 and 11 can be found in Supplementary Data 1 and 2, respectively. The raw image input data from this study can be available by the corresponding author (I.C.) upon reasonable request.

## Code availability
The MAxSIM height reconstruction code and the Graphical User Interface (GUI) are available at https://github.com/ichung-lab/maxsim.

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

## Acknowledgements

I.C. is supported by an NSF 1945373 grant, GWCC and Katzen Research Cancer Research Pilot Award, and GW Technology Maturation Award. We thank L. Shao for the helpful discussion, K. Yun for managing cells, P. Nalamalapu for creating the GitHub page, D. Chaterjee for confirming the functionality of our code, and B. Coblitz for discussing naming the technologies.

## Author contributions

I.C. conceived and supervised the project, secured funding, and designed the hardware and software. P.F.G.R. and I.C. built the MAxSIM set-up and performed experiments. P.F.G.R. wrote the code and draft of the readme file. Y.L. fabricated the HCM. I.C., S.P., A.A., and M.D. designed biological samples; A.A., T.V., H.J., and H.W.S. prepared those samples. I.C. wrote the manuscript with comments from all authors.

## Competing interests

I.C. submitted a patent application on the technologies related to the device (height-controlled mirror) developed in this work. All other authors declare on competing interests.

## Inclusion & Ethics statement

Primary neurons were derived from rat embryos in accordance with relevant guidelines and regulations and approved by the Institutional Animal Care & Use Committee at Thomas Jefferson University. Human tonsils, used in the study, were from patients at Children's National Medical Center, Washington, DC, and were exempt from NIH Institutional Review Board review, as per the Office of Human Research Protections. Sample donor demographics were not collected.
