## [Peer Review File · Communications Biology]

Reviewers' comments:

Reviewer #1 (Remarks to the Author):

This manuscript by Gardeazabal Rodriguez and colleagues presents an elegant multi-angle crossing structure illumination microscopy technique that allows for precision axial localization. The manuscript is well written and the method presented allows to obtain 3D topological mapping of live cells' structures. Here are my comments/questions:

1. The authors used MAXSIM to map the nanoscale 3D topology of the plasma membrane in different cell types.

- Differently from the experiments with IgG1-Alexa 488-conjugated-spin coated glass substrates where they used HCM ridge $h_0 = 10.7\mu\text{m}$, when probing the cells, they chose an HCM ridge height of $22\mu\text{m}$. How should the HCM height be chosen?
- Quantification of multiple repetitions and statistical analysis for each of the biological example should be provided.
- For Naïve -B cells, was the PM measured on the planar lipid bilayer?

2. In the SIM setup the authors report a 63x objective with 1.2 NA, can the authors comment on the numerical aperture and the minimal requirement of the system?

3. The authors of this manuscript use a minimum SiO₂ height of $1\mu\text{m}$, can they elaborate on the use of height between 500nm and $1\mu\text{m}$? This range can be found in various article using SAIM.

4. About the h , the authors report that " $h < 1000\text{nm}$ empirically leads to poor reconstruction when using a Si mirror covered with a $1\text{-}\mu\text{m}$ -thick SiO₂ layer". It would be interesting to see their reconstruction in the supplementary data.

5. Line 79-82: Published examples of use of SAIM include mapping of focal adhesion and adherens junctions in cells seeded on biomimetic substrates (without unroofing of the cell) by P. Kanchanawong's group.

6. Line 221: the authors report a Python-based height reconstruction code: please include the link to GitHub for easy access.

Reviewer #2 (Remarks to the Author):

My first opinion of the manuscript by Gardeazabal Rodriguez et al. is positive. This document presents a new approach, multi-angle-crossing structure illumination microscopy (MAXSIM), with in vivo 3D-topological mapping capability of the cellular plasma membrane.

The document is technically sound, experimental and theoretically well-documented. The extended document provides valuable information to illustrate the power of this new technique.

The following are small suggestions to improve the already high quality of the manuscript.

1- Image quality, size, and organization of the figures do not show the details of rugosity or z-axis depth in a graphical perspective. I suggest playing with ROIs to offer a zoom-in for specific regions. Also, the organization of the figures can be improved.

2- Date availability to test their method, authors claim for data and code viability at their GitHub. However, it wasn't accessible until login or request.

3- Examples to show resolution, the examples used in the manuscript aim to show the topography of the plasma membrane, and it ok. But, I believe it is necessary to have a figure showing the resolution of the method (x/y and z) compared with other approaches.

Reviewer #3 (Remarks to the Author):

Rodriguez et al reports a new scanning angle interference microscopy (SAIM) technique that places SiO₂/Si mirror on the top of the samples and combines with 2D-SIM. The approach is interesting and it may have some impacts on the study of plasma membranes. However, there are several concerns in

the manuscript so I cannot support its publication.

Major comments:

1. Experimental evidence is lacking on the measured height for well-known reference samples, for example, beads or microtubules.
2. In Extended Data Fig 7, the authors show AF488-IgG result but it's hard to believe that the thickness of spin coated surface has 1.2 μm variation in height. I don't think this is not a good proof-of-principle experiment.
3. Similarly, in Fig.2, why does MCF7 cell show $\sim 6 \mu\text{m}$ variation in height on the surface?
4. The frame rate of SLM is usually much slower than a galvo scanner. So the authors' claim regarding fast optoelectronic control for incidence angle is inaccurate.
5. Please show calibration data of incidence angle over SLM pattern.
6. It was unclear how to reuse HCM, especially after live-cell imaging. Many biomolecules should nonspecifically bind to the surface of SiO_2 which will degrade reflection.
7. If the authors want to show improvement of the performance of their technique over other methods, they should show "experimental" comparison results. For example, how much would this method be better than 3D-SIM or 3D-STORM?
8. As many frames are required to reconstruct M \times SIM, photobleaching would lead to reduce the fluorescence intensity. Please explain how to manage this.

Minor comments:

1. Line 56-58: I don't think this statement is correct. "However, most of these approaches can be experimentally challenging due to cumbersome sample placement geometry and long data collection times."
2. Line 74: Ref37 reported the imaging speed of 0.3 Hz.
3. Line 75: Explanation of the low imaging speed of SAIM is not correct. It is due to the number of incidence angles like the authors' approach.

Responses to Reviewers' Comments:

We are grateful for all the referees' comments aimed at improving our manuscript. We have diligently addressed every comment they raised and are pleased to note that the manuscript has been significantly improved. Please find the point-by-point responses below.

Reviewers' comments:

Reviewer #1 (Remarks to the Author):

This manuscript by Gardeazabal Rodriguez and colleagues presents an elegant multi-angle crossing structure illumination microscopy technique that allows for precision axial localization. The manuscript is well written and the method presented allows to obtain 3D topological mapping of live cells' structures. Here are my comments/questions:

We sincerely thank the reviewer for the positive evaluation and valuable suggestions for improving the manuscript. Below, we have diligently addressed the comments, as demonstrated in the revised version.

1. The authors used M_AXSIM to map the nanoscale 3D topology of the plasma membrane in different cell types.

• Differently from the experiments with IgG1-Alexa 488-conjugated-spin coated glass substrates where they used HCM ridge $h_0 = 10.7 \mu\text{m}$, when probing the cells, they chose an HCM ridge height of $22 \mu\text{m}$. How should the HCM height be chosen?

We appreciate the reviewer for pointing this out. We selected the HCM height based on the dimensions of the objects under study. Considering that the IgG1-Alexa conjugates were spin-coated, we anticipated minimal height fluctuation compared to the cell surface, which features various structures of varying heights. Consequently, we opted to use a low-height ridge available ($10.7 \mu\text{m}$) during the experiment. However, electron microscopy images revealed that SKBR3 cells can reach heights of up to $\sim 20 \mu\text{m}$. To ensure that the mirror was positioned above all cells, we employed an HCM with taller ridges ($22 \mu\text{m}$ high).

In the manuscript, we emphasized the importance of using an HCM taller than the cell height to ensure proper positioning of the mirror above the cells during measurements (lines 223-225).

“We chose an HCM ridge height of $22 \mu\text{m}$ to cover both MCF7 and SKBR3 cells seeded on the bottom glass substrate. We recommend to use a HCM that is taller than the cell height to ensure that the mirror is positioned above the measured cells.”

• Quantification of multiple repetitions and statistical analysis for each of the biological example should be provided.

Thank you for this valuable suggestion. To address this, we compared the height distributions of flat (MCF7) and deformed (SKBR3) cell membranes, as well as the Alexa dye – IgG1 conjugates spin-coated glass surface ($n = 3$). However, the distributions of naive B cells and GC cells exhibited significant heterogeneity within each group. This heterogeneity made it challenging to average their behaviors due to the limited number of cell samples. For naive B and GC cells, therefore, we decided to use the Gaussian widths of the only data set shown in Fig. 2e-f. The analysis showed that the average Gaussian width of the height distribution of MCF7 cells (width:

493 ± 315 nm) was narrower than that of SKBR3 cells (width: 1009 ± 594 nm). The width of the Alexa dye – IgG1 spin-coated glasses was the narrowest (width: 77 ± 35 nm). These results are described in Supplementary Figure 11 and the main text.

“Supplementary Fig. 11. Height distribution plots for the fitted heights with NELD < 0.1 are shown for independent MxSIM imaging of Alexa 488-IgG1 conjugates spin-coated on a glass surface (a; represented by eggplant bar graphs), and WGA-Alexa 555-stained MCF7 (b; represented by light green bar graphs) and SKBR3 (c; represented by orange bar graphs) cells. d. The mean and standard deviation values of the Gaussian widths of the distributions (a-c) are plotted for the three systems, showing a significantly narrow height distribution (width: 77 ± 35 nm) for Alexa 488-IgG1, compared to MCF7 (width: 493 ± 315 nm) and SKBR3 (width: 1009 ± 594 nm). As predicted, the membrane height distribution was wider in SKBR3 than in MCF7 cells.”

Lines 188-190

“As expected, the overall height distribution from three independent experiments display a narrow axial distribution with a Gaussian width of 77 ± 35 nm (mean ± standard deviation), as shown in Supplementary Fig. 11.”

Lines 225-228

“We compared the Gaussian widths of the height distributions for three MCF7 cell (width: 784 nm) and the SKBR3 cell (width: 1410 nm). The average widths for MCF7 and SKBR3 are 493 ± 315 nm (n=3) and 1009 ± 594 nm (n=3), as predicted (Supplementary Fig. 11).”

Lines 235-238

“Consistent with this observation, the basal surface of naïve B cells (stained for BCRs) was smoother (Gaussian width: 427 nm; Fig. 2e) than the basal surface of GC B cells (1,702 nm; Fig. 2f), which showed protrusions that facilitate antigen-driven selection as part of the immune response, enhancing affinity discrimination of antigen³⁹.”

• *For Naïve -B cells, was the PM measured on the planar lipid bilayer?*

We appreciate the reviewer’s question. Indeed, the experiment was conducted on a planar lipid bilayer coated with antigens. A comprehensive description of this sample preparation is provided in the method section of our manuscript as below.

Lines 442-472

“1. Tonsillar B cell isolation and preparation of B cell subpopulations:

Fresh human tonsils were obtained from patients undergoing tonsillectomies at the pathology department of the Children’s National Medical Center, Washington, DC. The use of these tonsils for this study was exempted from review by the NIH Institutional Review Board, following the guidelines issued by the Office of Human Research Protections. Tonsil cells were sorted into naïve B cells and GC B cells based on the expression of CD19, IgD, and CD10 as follows: naïve: CD19+, IgD+, CD10-; GC: CD19+, IgD-, CD10+.

2. Surface BCR labeling and activation on antigen-bound PLB:

Sorted cells were incubated in 5% CO₂ at 37°C for 2 hours. After incubation, cells were labeled for BCRs using Fab fragments of antibodies against IgG or IgM heavy chain (109-586-129; 109-587-043, Jackson Immunoresearch lab). Cells were then activated for 8 minutes at 37°C using antigens attached to the planar lipid bilayer prepared as previously described⁵¹.

For the preparation of the antigen-bound planar lipid bilayer, biotin-containing small unilamellar vesicles (SUVs) were prepared by mixing a 100:1 molar ratio of 1,2-Dioleoyl-sn-Glycero-3-phosphocholine (DOPC) and 1,2-Dioleoyl-sn-Glycero-3-phosphoethanolamine-cap-biotin (DOPE-cap-biotin) (Avanti Polar Lipids, Alabaster, AL), sonicating, and resuspending in phosphate-buffered saline (PBS) at a lipid concentration of 5 mM. Aggregated liposomes were cleared by ultracentrifugation and filtering.

Planar lipid bilayers (PLBs) were formed in Lab-Tek Chambers (Nalge Nunc, Rochester, NY), with the coverglasses replaced with nanostrip-washed coverslips. The coverslips were incubated with 0.1 μ M SUVs in PBS for 10 minutes. After washing with 20 ml PBS, the bilayer was incubated with 2.5 mg/ml of streptavidin for 10 minutes, and excess streptavidin was removed by washing with 20 ml PBS. The bilayers were further incubated for 20 minutes with 10 nM biotinylated pAb against human Igk light chain as a surrogate for the antigen. The quality of PLBs and the mobility of antigens in the lipid bilayers were confirmed by analyzing the proteins labeled with fluorescent dyes. Activated cells were fixed with 4% paraformaldehyde for 10 minutes at room temperature, washed with PBS, and then chambers were packed with parafilm for courier.”

2. In the SIM setup the authors report a 63x objective with 1.2 NA, can the authors comment on the numerical aperture and the minimal requirement of the system?

Thank you for raising this important point. Indeed, we tested different objectives to determine the desired qualifications for M_{Ax}SIM. These objectives varied in working distances (WD = 0.15 - 0.4 mm at D = 0.17 mm), immersion media (water and glycerin), and numerical aperture (NA = 1.15-1.3). After a thorough evaluation of the incident angles within the desired scan angle range using various SLM patterns, we identified that a 63x water objective with 1.2 NA (Objective C-Apochromat 63x/1.20 W Corr M27 WD = 0.28 mm at D = 0.17 mm) was most suitable for M_{Ax}SIM. This specific objective provided a large excitation scan range and sufficient photon collection for performing M_{Ax}SIM.

While a glycerin objective with similar capabilities (63x, NA = 1.3, WD = 0.17 mm at D = 0.17 mm) could also be used, we ultimately chose the water immersion objective. This decision was based on the observation that glycerin sometimes failed to uniformly cover the objective tip, leading to non-reproducible incident angles.

In summary, the minimum requirements for an objective to perform M_{Ax}SIM are 63x, NA = 1.2 (water), and WD > 0.15 mm. We have included the above information in the method section.

Lines 411-424:

“We tested different objectives, each varying in working distances (WD = 0.15-0.4 mm at D = 0.17 mm), immersion media (water and glycerin), and numerical aperture (NA = 1.15-1.3). After thorough evaluation, we identified that a 63x water objective with 1.2 NA (Objective C-Apochromat 63x/1.20 W Corr M27 WD=0.28 mm at D=0.17 mm) was the most suitable for M_{Ax}SIM. This particular objective allowed for a large excitation scan range and provided sufficient photon collection for our purposes. Although a glycerin objective with similar capabilities (63x, NA = 1.3, WD = 0.17 mm at D = 0.17 mm) could also be used, we ultimately chose the water immersion objective, because of our observation that glycerin sometimes failed to uniformly cover the objective tip, leading to non-reproducible incident angles. To ensure successful implementation of M_{Ax}SIM, it is crucial to carefully select an objective by evaluating the incident angles in the desired scan range using different SLM patterns. The minimum requirements for an objective to perform M_{Ax}SIM are 63x, NA = 1.2 (water), and WD > 0.15 mm.”

3. *The authors of this manuscript use a minimum SiO₂ height of 1 μm, can they elaborate on the use of height between 500nm and 1 μm? This range can be found in various article using SAIM.*

We appreciate the reviewer for pointing this out. We have observed that the fidelity of height reconstruction depends on having a sufficient number of fringes generated for a specific probe height and the SiO₂ thickness. To demonstrate this relationship, we simulated the excitation intensity fringes at a wavelength of 488 nm for three commercially available SiO₂ thicknesses: 0.5, 1, and 10 μm (Supplementary Figure 7). For our analysis, we maintained a fixed height of the probe at 1, 10, and 20 μm.

Upon comparing the results, we observed that the difference in fringe numbers between 0.5 μm and 1 μm is very small, whereas 10 μm creates significantly more fringes, which could potentially pose challenges for topological mapping of the apical membranes of tall cells. As a result, both 0.5 μm and 1 μm SiO₂ thicknesses can be considered viable options for cells positioned on the mirror. These findings are discussed in the main text.

Lines 145-149:

“The number of interference fringes depends on the SiO₂ thickness as well as the height of a chromophore, as demonstrated by our simulated data (Supplementary Fig. 7). Different SiO₂ thicknesses such as 500 nm³⁸ or 10 μm (which are commercially available) yield similar (in the 500 nm case) or more excitation interference fringes than for the 1 μm thick SiO₂ case³⁴.”

4. *About the h, the authors report that “h < 1000nm empirically leads to poor reconstruction when using a Si mirror covered with a 1-um-thick SiO₂ layer”. It would be interesting to see their reconstruction in the supplementary data.*

We thank the reviewer for raising this point and we agree that it is worthwhile to compare *the h < 1000nm case with the h > 5000 nm case*. We have implemented this suggestion in various sections of the manuscript (Lines 149-156; Supplementary data 8,9) to enable a comparison of the reconstruction fidelity using 100 nm beads placed on a glass substrate (*h < 1000 nm case*) versus on a SiO₂/Si mirror with a 7 μm ridge height (*h > 5000 nm case*). As demonstrated, the standard least-square fit offered by Python failed to fit the experimental data accurately (height found by least square fitting: 2.8 μm). While our software can determine the optimal initial parameters for the least square fitting, it was specifically designed to optimally fit experimental data with multiple fringes. When using the auto-generated initial parameters through the least square fitting tool in Python, the fitting results were poor (height found by our algorithm: 43 μm). However, for the beads on the bottom glass using an HCM, the fitting fidelity was superior to the case where beads were placed on the mirror (Supplementary Figure 8) as described in Supplementary Figure 9.

Lines 149-156:

“An adequate number of modulation fringes within an incident angle range θ was crucial to yield high-fidelity height reconstruction. For instance, $h < 1,000$ nm empirically leads to poor reconstruction when using a Si mirror covered with a 1-μm-thick SiO₂ layer, as demonstrated by the example raw data and fitted curves for 100 nm microspheres placed on a SiO₂/Si mirror (Supplementary Fig. 8). A chromophore distance >5 μm away from the 1-μm-thick SiO₂-covered Si mirror creates more fluorescence interference fringes and enables higher-fidelity height reconstruction (Supplementary Fig. 9), compared to shorter distances (<1 μm).”

5. Line 79-82: *Published examples of use of SAIM include mapping of focal adhesion and adherens junctions in cells seeded on biomimetic substrates (without unroofing of the cell) by P. Kanchanawong's group.*

We thank the reviewer for pointing this out. We have included various references, including the one suggested by the reviewer, to showcase the past application areas of SAIM.

Lines 69-70:

“Lateral imaging is diffraction-limited and most SAIM applications have been primarily used to map the topology of the basal cell surface, focal adhesion sites, and cytoskeletons underneath the basal cell surface that was adhered to the SiO₂/Si mirror^{19,21-25}.”

6. Line 221: *the authors report a Python-based height reconstruction code: please include the link to GitHub for easy access.*

We appreciate this suggestion. We have made modifications to the code acquisition process to download the code directly from our GitHub site.

Reviewer #2 (Remarks to the Author):

My first opinion of the manuscript by Gardeazabal Rodriguez et al. is positive. This document presents a new approach, multi-angle-crossing structure illumination microscopy (MAxSIM), with in vivo 3D-topological mapping capability of the cellular plasma membrane. The document is technically sound, experimental and theoretically well-documented. The extended document provides valuable information to illustrate the power of this new technique. The following are small suggestions to improve the already high quality of the manuscript.

We sincerely thank the reviewer for the positive evaluation and valuable suggestions to improve the manuscript. Please find the point-by-point responses below.

1- Image quality, size, and organization of the figures do not show the details of rugosity or z-axis depth in a graphical perspective. I suggest playing with ROIs to offer a zoom-in for specific regions. Also, the organization of the figures can be improved.

We appreciate the reviewer for raising this important point. In response to this comment, we have made several adjustments to enhance the clarity of our visualizations of cell membrane topology. In the previous version, we used height ranges covering both “flat” and “deformed” cell membranes. Therefore the “flat” membrane cells appeared with a single-color tone whereas the “deformed” membranes exhibited a wider color range. This scheme made it difficult to visualize the membrane features of “flat” cells. In addition, we included all pixels regardless of the fitting fidelity values. To rectify this problem in the revised manuscript, we rescaled the height range to exclude the height values resulting from poor fitting ($NELD > 0.2$) for each cell, which improved the visualization of membrane features in cells. Furthermore, as suggested by the reviewer, we have provided zoomed-in images for specific regions of interest. All these improvements are now incorporated into Figure 2-4.

2- Date availability to test their method, authors claim for data and code viability at their GitHub. However, it wasn't accessible until login or request.

We thank the reviewer for this valuable suggestion. We have made modifications to the code acquisition process to download the code directly from our GitHub site.

3- Examples to show resolution, the examples used in the manuscript aim to show the topography of the plasma membrane, and it ok. But, I believe it is necessary to have a figure showing the resolution of the method (x/y and z) compared with other approaches.

We thank the reviewer for the helpful recommendation. We have incorporated a table (Supplementary Table 1) summarizing the resolution and localization accuracy of other relevant 3D super-resolution imaging (3D SIM) or 3D localization approaches (3D-STORM, 3D-PALM, 3D-STED, iPALM, and SAIM) in comparison to theoretical accuracy and lateral resolution of M_AXSIM.

Reviewer #3 (Remarks to the Author):

Rodriguez et al reports a new scanning angle interference microscopy (SAIM) technique that places SiO₂/Si mirror on the top of the samples and combines with 2D-SIM. The approach is interesting and it may have some impacts on the study of plasma membranes. However, there are several concerns in the manuscript so I cannot support its publication.

We sincerely thank the reviewer for raising important points, and we have made diligent efforts to address them. Please find the point-to-point responses below.

Major comments:

1. Experimental evidence is lacking on the measured height for well-known reference samples, for example, beads or microtubules.

We very much appreciate the reviewer for raising this important point. First of all, we would like to point out that we are measuring a relatively long distance between the object of interest on the bottom glass surface and the top reflecting mirror surface. Due to the height of cells, however, the distance between the mirror and bottom surface should be larger than 10 μm . Since our theoretical localization accuracy is proportional to this distance, namely 0.7% of the height, the localization uncertainty for the 10 μm height is 70 nm. Given this localization accuracy, both a 100 nm microsphere or 24 nm thick microtubules would not serve as effective reference systems for M_AXSIM. Consequently, we decided to measure the relative height differences on the dye-coated glass surface as our reference system instead.

2. In Extended Data Fig 7, the authors show AF488-IgG result but it's hard to believe that the thickness of spin coated surface has 1.2 μm variation in height. I don't think this is not a good proof-of-principle experiment.

3. Similarly, in Fig.2, why does MCF7 cell show ~6 μm variation in height on the surface?

We thank the reviewer for raising these concerns. As comments #2 and #3 are related, we will address them together.

Regarding the reviewer's comment #2, we explain why the measurement of Alexa dye-covered glass surface can be a successful reference measurement. We understand the reviewer's concern about the broad height ranges observed for the Alexa488-IgG coated glass surface (1.2 μm) in the previous representation, which didn't make this measurement an efficient reference. However, the large height range was due to the inclusion of poorly fitted pixel heights into the color bar range. In reality, the height distribution for the Alexa488-IgG coated glass surface was narrow (Gaussian width = 77 ± 35 nm) as shown in Supplementary Data 10, which is significantly narrower height distribution of the plasma membranes of "flat" cells (Gaussian width = 493 ± 315 nm).

nm). Therefore, Alexa dye-coated coverslip serve as an efficient reference measurement for M_AxSIM.

Regarding comment #3, in the previous manuscript, we used a wide height range to cover entire height ranges of both flat and deformed cells, resulting that the height reconstruction images for "flat" membrane cells mostly exhibited a single color tone. The reasons for the wide color ranges were twofold: i) to compare cell surfaces with allegedly flat membranes versus deformed membranes, as reported in our previous work (MCF7 versus SKBR3 cells) and by Dr. Pierce (naïve B cells versus germinal center B cells), and ii) to include the heights of all pixels regardless of some being associated with poor fitting (NELD > 0.2) in the color bar ranges.

We acknowledge that the wide height ranges may have caused confusion. To address this concern, we have rescaled the color bar to include only heights associated with small NELD (< 0.2) for each cell. This adjustment allows for more effective visualization of the membrane features of cells, also addressing Reviewer 2's comment #1/3 regarding the improvement of height-reconstructed image visualization.

4. The frame rate of SLM is usually much slower than a galvo scanner. So the authors' claim regarding fast optoelectronic control for incidence angle is inaccurate.

We appreciate the reviewer for pointing this out, and we acknowledge this point. Galvo is indeed faster than SLM in switching, but it does not offer the flexibility of generating user-defined illumination patterns, which is necessary for performing M_AxSIM. Therefore, we have removed the previous sentence that compared the speeds between the two modalities.

5. Please show calibration data of incidence angle over SLM pattern.

We thank this reviewer for this helpful suggestion. We provided the calibration curve between the SLM pattern and the incident angle in the Supplementary Figure 3 and mentioned this in the main text.

Lines 118-120:

"Our calibration data demonstrates excellent accuracy, as the incident angles are within a remarkable 2% margin of error compared to the theoretical angles. (Supplementary Fig. 3)."

6. It was unclear how to reuse HCM, especially after live-cell imaging. Many biomolecules should nonspecifically bind to the surface of SiO₂ which will degrade reflection.

We appreciate the reviewer for bringing up this concern. In fact, we have been using the same HCM for over 2 years, employing the typical cleaning procedure with isopropanol. The photoresist we use, SU-8, is known for its remarkable resistance, as indicated by the manufacturer's webpage (<https://kayakuam.com/products/su-8-series-and-kmpr-plasma-removal-rework/>), which states, "removal or reworking highly crosslinked SU-8 epoxy resist is difficult because it is chemically very stable. To rework very thick SU-8 layers economically without damaging other microstructures requires a highly selective process." We have now incorporated the cleaning procedure and a brief description of the resistance of SU-8 in the method section.

Lines 82-84:

"The HCM is reusable, thus saving the time and costs required for fabrication (See the "HCM cleaning procedure" in the method section)"

Lines 400-406:

“HCM cleaning procedure

The HCM is reusable if the following cleaning steps are adhered to. 1. Dunk the used HCM in fresh 100% isopropanol for 10 min. 2. Rinse it with deionized water for 2 min. 3. Air dry the HCM. 4. Store it in a dust-free storage box. Our group has used the same HCM for over 2 years. The photoresist we use, SU-8, is known for its remarkable resistance⁶⁰, which contributes to the durability of our HCM.”

7. If the authors want to show improvement of the performance of their technique over other methods, they should show "experimental" comparison results. For example, how much would this method be better than 3D-SIM or 3D-STORM?

We thank the reviewer for this suggestion. This manuscript does not aim to claim that M_AxSIM is superior to 3D-SIM or 3D-STORM. Instead, we present M_AxSIM as an add-on system to 2D-SIM, enabling 3D axial localization. Our intention is to avoid portraying M_AxSIM as a method that surpasses existing super-resolution and localization techniques. Nevertheless, we created a table as shown in Supplementary Table 1 to provide a comprehensive comparison of axial localization and resolution between the techniques and we revised the introduction of the manuscript accordingly.

Lines 317-320:

“Supplementary Table 1 summarizes the localization accuracy of 3D localization microscopy methods and the resolution of 3D super-resolution imaging techniques in comparison to the theoretical axial accuracy and lateral resolution of M_AxSIM.”

8. As many frames are required to reconstruct M_AxSIM, photobleaching would lead to reduce the fluorescence intensity. Please explain how to manage this.

Thank you for this helpful comment. Currently, our live-cell imaging using M_AxSIM with Alexa dyes is feasible with a 50 ms exposure time for ~ 30 s, as determined by our NELD assessment of the localization (see Supplementary Figure 15). However, we agree with the reviewer that for longer imaging sessions and/or shorter exposure times, utilizing more photostable and brighter dyes, such as Janelia Fluor® dyes, would be advantageous. We have addressed this consideration in the manuscript.

Lines 288-291:

“Future efforts should be made to further improve time resolution while maintaining excellent fitting fidelity by utilizing more photostable and brighter chromophores, such as Janelia Farm dyes⁴², and fine-tuning SiO₂ thickness for live-cell imaging.”

Minor comments:

1. Line 56-58: I don't think this statement is correct. "However, most of these approaches can be experimentally challenging due to cumbersome sample placement geometry and long data collection times."

2. Line 74: Ref³⁷ reported the imaging speed of 0.3 Hz.

3. Line 75: Explanation of the low imaging speed of SAIM is not correct. It is due to the number of incidence angles like the authors' approach.

We appreciate these valuable comments. Our introduction has been significantly rewritten to reflect the points raised by the reviewer.

We addressed the reviewer's points as shown below:

1. We agree with the reviewer and removed the sentence.
2. This is mentioned in lines 64-65.
3. We agree with the reviewer that the scanning angle requirement contributes to the low time resolution of SAIM or MAXSIM. To avoid any confusion, we have chosen to remove the mentioned sentence from the new introduction.

REVIEWERS' COMMENTS:

Reviewer #1 (Remarks to the Author):

Thanks to the author for their point by point reply. While I am quite happy with most of their replies, I am still not sure about the applicability of the method to biological samples as demonstrated by the high variability in their results.

I would suggest to measure well-known structures (maybe focal adhesion proteins in epithelial cells?) with less variability than the membrane of naive B cells and GC cells which exhibited significant high heterogeneity within such a small group (n=3) and thus does not help in proving the suitability of your method to a biological sample.

Reviewer #2 (Remarks to the Author):

I am satisfied with the work done in the manuscript. The document has improved significantly.

Reviewer #3 (Remarks to the Author):

The authors have clearly made an effort to improve the manuscript and the revised version is substantially stronger than the original manuscript. No further revision would be needed except Supplementary Table 1. All the localization accuracy (3D-STORM, PALM, STED and iPALM) need to be moved to spatial resolution. For example, localization accuracy of 3D-STORM (xy) is a few nm.

Responses to Reviewers' Comments:

We sincerely appreciate the referees' positive feedback and their satisfaction with the recent revisions we made to the manuscript. Below, please find our responses to their comments.

REVIEWERS' COMMENTS:

Reviewer #1 (Remarks to the Author):

Thanks to the author for their point by point reply. While I am quite happy with most of their replies, I am still not sure about the applicability of the method to biological samples as demonstrated by the high variability in their results.

I would suggest to measure well-known structures (maybe focal adhesion proteins in epithelial cells?) with less variability than the membrane of naïve B cells and GC cells which exhibited significant high heterogeneity within such a small group (n=3) and thus does not help in proving the suitability of your method to a biological sample.

We sincerely thank the reviewer for the positive evaluation and valuable suggestions. We fully agree that measuring the heights of focal adhesions from the mirror could lead to more uniform height distributions than direct measurements of membrane heights. Nevertheless, the aim of our experiment was to distinguish different height distribution patterns from diverse systems, anticipating inherent differences. We believe we achieved this by presenting differential height distributions of three distinct systems: dye-antibody coated glass, and MCF7 and SKBR3 cells on glass. The Gaussian widths were 1009 ± 594 nm for SKBR3 cells, 493 ± 315 nm for MCF7, and 77 ± 35 nm for Alexa 488-IgG1 (Supplementary Fig. 11). As expected, the widths of the height distribution were the largest for HER2-oncoprotein-enriched SKBR3 cells and smallest for the dye-antibody planar coat. These relative differences were in line with our earlier observations regarding the HER2-level-dependent cell-surface morphologies between MCF7 and SKBR3 cells (Chung et al., Nature Communications 7, 12742 (2016)). Therefore, we do not anticipate that introducing data of another flat object, such as focal adhesion sites, might not offer significant additional insights into our study.

We believe that differences in height distribution across various systems will become more apparent when they exceed the heterogeneity observed within a single system. This was evident in our comparison of height distributions between naïve B cells and GC B cells. Given that these cells were placed on antigen-coated artificial lipid bilayers rather than the bare glass substrates used for MCF7 and SKBR3 cells, we hypothesize that the observed large heterogeneity arises primarily from variations in local antigen concentrations. As MxSIM measurements are performed at the single-cell level, it's plausible that some cells encountered higher antigen concentrations, while others detected minimal levels. We even anticipate such variability within individual cells. We discussed the potential sources of this inherent heterogeneity in lines 246-248 of the main text.

"We noted heterogeneity in the plasma membrane morphologies and height distributions of both naïve B and GC B cells, potentially due to variations in local antigen densities on the lipid bilayer. Therefore, we did not conduct statistical comparisons."

Reviewer #2 (Remarks to the Author):

I am satisfied with the work done in the manuscript. The document has improved significantly.

We very much appreciate the reviewer's positive acknowledgment of the updates we made to the manuscript, following the invaluable suggestions this reviewer offered.

Reviewer #3 (Remarks to the Author):

The authors have clearly made an effort to improve the manuscript and the revised version is substantially stronger than the original manuscript. No further revision would be needed except Supplementary Table 1. All the localization accuracy (3D-STORM, PALM, STED and iPALM) need to be moved to spatial resolution. For example, localization accuracy of 3D-STORM (xy) is a few nm.

We are genuinely grateful to the reviewer for their positive feedback on the revisions made to our manuscript and for pointing out the need to revise Supplementary Table 1. In response, we have updated the table, which is now available in the Supplementary Information File. We again thank this reviewer for making this manuscript much stronger than the previous version.